# Microelement Variability in Plants as an Effect of Sewage Sludge Compost Application Assessed by Different Statistical Methods

**Monika Jakubus [1],\* and Małgorzata Graczyk [2]**

[1]   Department of Soil Science and Land Protection, Poznan University of Life Sciences,
     ul. Szydłowska 50, 60–656 Poznań, Poland
[2]   Department of Mathematical and Statistical Methods, Poznan University of Life Sciences,
     ul. Wojska Polskiego 28, 60-637 Poznań, Poland; malgorzata.graczyk@up.poznan.pl
**\***   Correspondence: monika.jakubus@up.poznan.pl

**Abstract:** This research deals with the effects of soil amendment with sewage sludge compost (SSC) on the accumulation of microelements (Cu, Zn, Mn, Ni, Fe) in plants—white mustard (*Sinapis alba*), triticale (*Triticale*) and white lupine (*Lupinus albus*)—cultivated on two contrasting soils (light vs. medium agronomic class). Additionally, the influence of experimental factors on variability of the harvest index (HI) was determined for individual plants and micronutrients. A 3-year pot experiment was conducted and SSC at the rate of 6 t·ha$^{-1}$ was applied into the soils. These changes were determined by ANOVA and subject to regression analysis and subsequently visualized. The study clearly demonstrated that SSC as an organic fertilizer had a significant, but weak effect on the microelements level variability in the shoots and grains of plants cultivated in crop rotation. Regardless of the experimental factors, on average, the plants accumulated higher amounts of Cu, Zn and Mn in the grains, and Fe and Ni in the shoots. Harvest index values confirmed the significance and variable translocation of microelements within plant organs. The influence of SSC on micronutrient contents in plant organs was more significant in the case of medium soil.

**Keywords:** sewage sludge compost; plant nutrition; agriculture; harvest index; three factor design; post-hoc analysis

## 1. Introduction

When considering proper plant nutrition generally, first of all, the macronutrient requirements need to be satisfied. However, current sustainable farming management recommendations also indicate micronutrient application needs. Micronutrients, for many years, have been considered as less essential in plant nutrition. Presently, it is well documented that micronutrients play an important role in increasing both the quantity and quality of crop yield through their involvement in the metabolism of N, P, K, Ca, Mg and S, as well as crop responses to environmental stress conditions [1]. A review conducted in [2] very exhaustively described not only the physiological roles of micronutrients as cofactors and participants of enzymes involved in metabolic processes, but also their functions in biotic and abiotic stress mitigation. Although microelements are required in very small quantities for normal plant growth and development, most crops are very susceptible both to micronutrient deficiency and excess [1]. Historically, micronutrients were applied only with natural organic fertilizers, such as manure, slurry, green manure or composts. Presently, more popular sources of micronutrients are artificial, mineral fertilizers. A wide range of such fertilizers are commercially available, so it is very easy to find an adequate product to satisfy plant demands. Unfortunately, these fertilizers are expensive. Moreover, currently, in relation to sustainable agriculture and environment protection principles,

farmers pay attention to the possibility of using organic fertilizers, which is in line with consumer expectations. Additionally, it should also be remembered that supplementation of micronutrients is not a routine practice, but it is performed when symptoms of deficiency appear on the plants. In such a case, despite the use of fertilizers, the micronutrients are very often provided to plants too late, causing low nutrient utilization, a reduction of yield and a deterioration of its quality [2]. Therefore, a more effective and pragmatic approach is connected with the systematic use of organic fertilizers, although they contain lower micronutrient amounts in comparison to concentrated mineral fertilizers. Additionally, organic fertilizers ensure a systematic release of nutrients, while maintaining their positive balance over a long period of time. Studies [3–5] indicate such a possibility. The cited authors proved that systematic long-term application of organic fertilizers not only results in an increase of soil organic matter, but also in an increase of micronutrient amounts available to plants. Sharma et al. [6] emphasize the significance of biosolid composting in terms of both their increasing mass and the rational use of nutrients and organic matter contained in them. Additionally, Thomas et al. [7] listed the positive aspects related to the utilization of recycling organic wastes as crop fertilizers. In this respect, an interesting proposal is an alternative to commonly used mineral fertilizers, which can be provided by compost prepared on the basis of sewage sludge. Sewage sludge is not only characterized by a higher content of organic matter and macronutrients, but also micronutrients [8]. Moreover, the use of composted sewage sludge is a very important strategy, complying with [9], and additionally meeting the main principles of sustainable agriculture and circular economy [4]. According to a circular economy philosophy, recyclable materials are reintroduced into the economy as new raw materials, thus increasing the security of supply. Such an approach was strengthened and supported by [10]. Moreover, Velten et al. [11] indicated the fact that sustainable agriculture must produce adequate amounts of high-quality food, protect resources and be both environmentally safe and profitable. To meet these challenges, innovative and new techniques also include some well-known measures, such as organic fertilization [12]. These rules are particularly important in relation to sewage sludge compost, because, as indicated by Sharma et al. [6], sewage sludge may be loaded with heavy metals, pesticides, insecticides, pharmaceuticals, detergents, personal care products or steroid hormones, and consequently, these compounds may be included in the food chain. Therefore, the utilization of sewage sludge compost must be kept under full environmental control, especially when considering agricultural uses.

Despite many positive aspects connected with compost application [4,13–15], some disadvantages should also be underlined. Obviously, slow mineralization and the resulting sluggish progress of nutrient release diminish the agricultural value of composts. However, in the case of micronutrients supplementation, slow liberation of nutrients from compost should be considered as a positive aspect of such fertilization, because during the vegetation period plants take up micronutrients in very small amounts at variable rates. Moreover, nutrient acquisition is governed by various factors, which must be considered during the entire process, starting from compost application through compost mineralization and nutrient losses, finally up to nutrient uptake. The most important factors, which play a significant role in micronutrient uptake are related to plant species (e.g., crop requirement, plant physiology) and soil conditions (e.g., soil reaction, soil moisture and organic matter contents).

Knowing that organic matter exhibits high availability to create stable bonds with most metals, one can expect low activity and limited influence of compost organic matter on the chemical composition of crops. Few studies have been dedicated to this problem, because it is difficult to enhance the direct effect of compost on plant chemical composition. It is much easier to determine the effect of compost on plant yield, as yield attributes are more evident and well documented. Both positive [16–19] and negative yield responses [20–23] have been reported for a wide variety of crops as a result of compost application to soils. Therefore, this study is focused on testing the following hypotheses: (1) sewage sludge compost (SSC) will serve as a valuable source of micronutrients for plants cultivated in crop rotation, which will be assessed based on the chemical composition of the shoots and grains as well as harvest index values; and (2) the soil type (light vs. medium agronomic class ) enriched with sewage

sludge compost will determine the micronutrient contents and harvest index variability assessed for individual plant species.

## 2. Material and Methods

### 2.1. Experimental Design

A 3-year (2013–2015) pot experiment was conducted at the experimental vegetation station at Swojec located near Wrocław (51°06′00″ N, 17°01′59″ E). Outdoor, natural conditions were provided for the experimental facility, because it was covered with wire mesh. The two soils used in the study: light (LS) and medium (MS), differed markedly in their properties. Samples were collected from a depth of 0–30 cm from an agricultural field. The light agronomic soil class (loamy sand) was classified as *Haplic Luvisols*, and the medium agronomic soil class (clay loam) was classified as *Haplic Cambisols*, according to WRB [24]. The sewage sludge compost (SSC) used for experiments met the requirements set out in the respective Polish law [25], and was produced for commercial purposes by a local composting facility. The compost was applied at a rate of 6 t·ha$^{-1}$ into the soils. Both soils and SSC were sieved through a 20 mm sieve before setting up the experiment. The total amounts of micronutrients in the soils and SSC were assessed according to the aqua regia procedure [26] and they are presented in Table 1. The used method is based on the soil sample or compost extraction with a mixture of concentrated nitric and hydrochloric acids (1:3) at a ratio of 1:7 at a high temperature (65–70 °C) run for 3 h.

**Table 1.** Total contents (mg·kg$^{-1}$) of micronutrients in sewage sludge compost and soils used in the experiment.

| Parameter | Sewage Sludge Compost (SSC) | Light Agronomic Soil Class (LS) | Medium Agronomic Soil Class (MS) |
|---|---|---|---|
| pH * | 6.8 | 6.5 | 7.0 |
| Mn | 346.9 | 38.3 | 261.6 |
| Cu | 156.6 | 5.1 | 14.3 |
| Zn | 569.9 | 20.3 | 42.3 |
| Fe | 10461 | 775.5 | 1335.7 |
| Ni | 35.8 | 8.1 | 25.6 |

* pH for sewage sludge compost was assessed in $H_2O$ and for soils in 1 M KCl (soil/compost: solution ratio = 1:2.5, w/v).

Dry soil samples of 10 kg were weighed in triplicate and they were thoroughly mixed with the dose of compost two weeks before plant cultivation. Each mixture was wetted to 60% field capacity. The compost was incorporated once, at the beginning of the experiment (before planting). The experiment was conducted in PVC pots (10 kg) in a randomized, factorial design with two soils (LS, MS), with one treatment with and the other without the SSC addition. As a result, the organization of the experiment included: T0—soil control (without compost addition) and T1—soil with the compost addition. Accordingly, the following combinations were in the experiment: LST0 (light soil without compost addition), LSTI (light soil with compost addition), MST0 (medium soil without compost addition) and MSTI (medium soil with compost addition). Three crops (as one of the experiment factors), i.e., white mustard (*Sinapis alba*) (WM), triticale (*Triticale*) (T) and white lupine (*Lupinus albus*) (WL), were planted in consecutive years (2013–2015) after harvesting in the same pot. The selection of crops was determined by their large-scale cultivation in Poland, due to typical soil and weather conditions. The experiment was conducted at a density of 10 plants per pot. The vegetation period of individual plants was typical of Polish conditions (from the end of March to the beginning of July for white mustard, from the end of March to the beginning of August for triticale and from beginning of April to the end of July for white lupine). Detailed data on weather conditions are included in the Supplementary Materials. The aboveground plant material was harvested at the maturation stage.

## 2.2. Analysis of Plant Materials

The aboveground plant material was divided into vegetative (shoot) and generative parts (grain with pods and ears). Plant material was dried at 60 °C, ground and ashed in a furnace (CZYLOK, FCF5SH) at 450 °C for 6 h. The ash was dissolved in 5 mL of 6 mol·dm$^3$ HCl and diluted to a constant volume with distilled water [27]. The obtained extracts were analyzed to assess Cu, Zn, Mn, Fe and Ni contents using atomic absorption spectrophotometry (ASA) in a Varian Spectra AA 220 FS apparatus. All the assays identifying the amounts of nutrients in the tested samples were performed in three replications.

Following Ma and Zheng [28], the nutrient harvest index (HI) of microelements was calculated as follows:

$$\text{Nutrient HI} = \frac{\text{nutrient content in grain}}{\text{nutrient content in shoots}} \tag{1}$$

## 2.3. Statistical Analysis

The following model of the experiment was adopted:

$$y_{ijkl} = m + a_i + b_j + c_k + ab_{ij} + ac_{ik} + bc_{jk} + abc_{ijk} + \varepsilon_{ijkl} \tag{2}$$

where $y_{ijkl}$ denotes the content of $l^{\text{th}}$ microelement in $k^{\text{th}}$ plant growing on $j^{\text{th}}$ soil with addition of $i^{\text{th}}$ compost. Here, $m$ is the general mean, $a_i$ - the main effect of the $i^{\text{th}}$ level of factor A (soil without or with compost), $b_j$- the main effect of the $j^{\text{th}}$ level of factor B (light or medium soil), $c_k$ - the main effect of the $k^{\text{th}}$ level of factor C (plant: white mustard, triticale and white lupine), $ab_{ij}$ - the effect of the interaction between the $i^{\text{th}}$ level of factor A and the $j^{\text{th}}$ level of factor B, $ac_{ik}$ - the effect of the interaction between the $i^{\text{th}}$ level of factor A and the $k^{\text{th}}$ level of factor C, $bc_{jk}$ - the effect of the interaction between the $j^{\text{th}}$ level of factor B and the $k^{\text{th}}$ level of factor C, $abc_{ijk}$ - the effect of the interaction between the $i^{\text{th}}$ level of factor A, the $j^{\text{th}}$ level of factor B and the $k^{\text{th}}$ level of factor C. Moreover, $\varepsilon_{ijkl}$ is the experimental error for $i, j, k, l$- unit, and it is the implementation of a random variable with a normal distribution with an expected value of zero and with the same variance. Within all the experimental units, the conditions are random, and the technical acquisition of observation of the Y feature is independent, i.e., the errors $\varepsilon_{ijkl}$ are uncorrelated. In order to determine how a response is affected by the levels of considered factors and to determine the cooperation between the factors three-way ANOVA was applied [29,30]. Tukey's procedure was used to establish homogeneous groups in order to compare considered factors: type of fertilization, soil and crop plant we used. The analysis was performed to test the basic hypothesis: factors A, B and C do not differentiate the levels of investigated properties, along with the hypothesis: there is no interaction between the levels of any two factors and between the three factors. Moreover, the linear regression analysis was performed. The obtained results are visualized in the form of boxplots. In Figures 1–3 boxplots for grains, shoots and HI are presented.

## 3. Results

In view of the hypotheses of this study, the effects of three factors—compost addition and its absence (T0 and T1), soil (LS, MS) and plant (WM, T, WL)—were studied and their interactions were analyzed. Generally, an increase was observed for all microelement contents in shoots of plants cultivated on the soil enriched with compost compared to those cultivated on the control soil (Table 2, Figure 1). Analyses of all the studied microelements showed their greater contents in the shoots of plants cultivated on medium soil, regardless of the compost addition. The highest contents of all elements were accumulated in triticale shoots (Figure 1).

Slightly smaller contents of Zn, Cu and Mn were observed in white mustard. The results presented in Table 2 indicate that the relationship between the compost and soil was statistically significant for Mn and Ni, and the greatest contents of these two elements were recorded in plants cultivated on medium soil enriched with SSC (Figure 1).

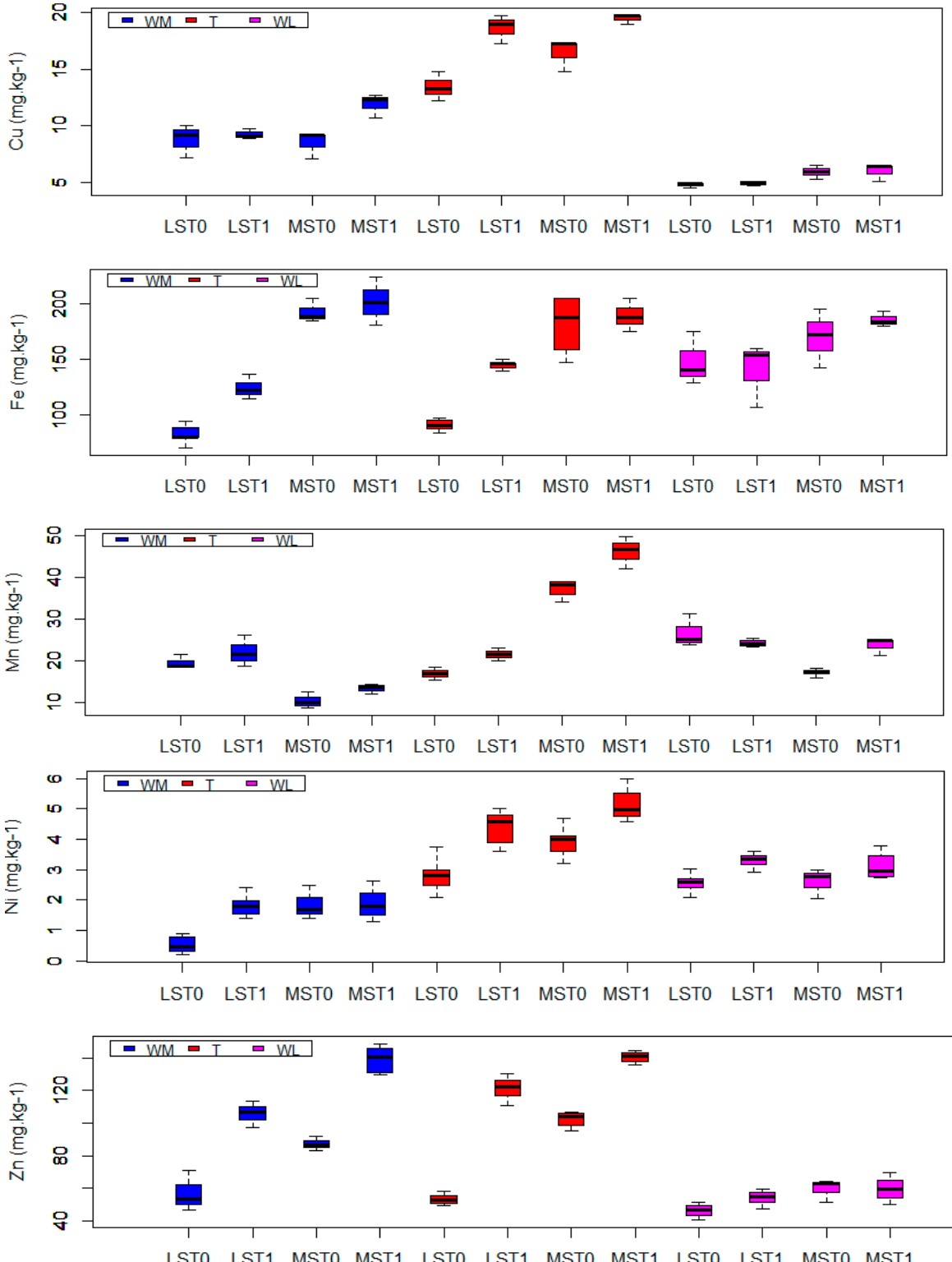

**Figure 1.** Content of microelements in shoots. In the figure, the minimal value, first quartile (Q1), median, third quartile (Q3), and the maximal value are given to display the distribution of data. On the OX axis, the combination of treatments are marked, for more details see description in Material and Methods.

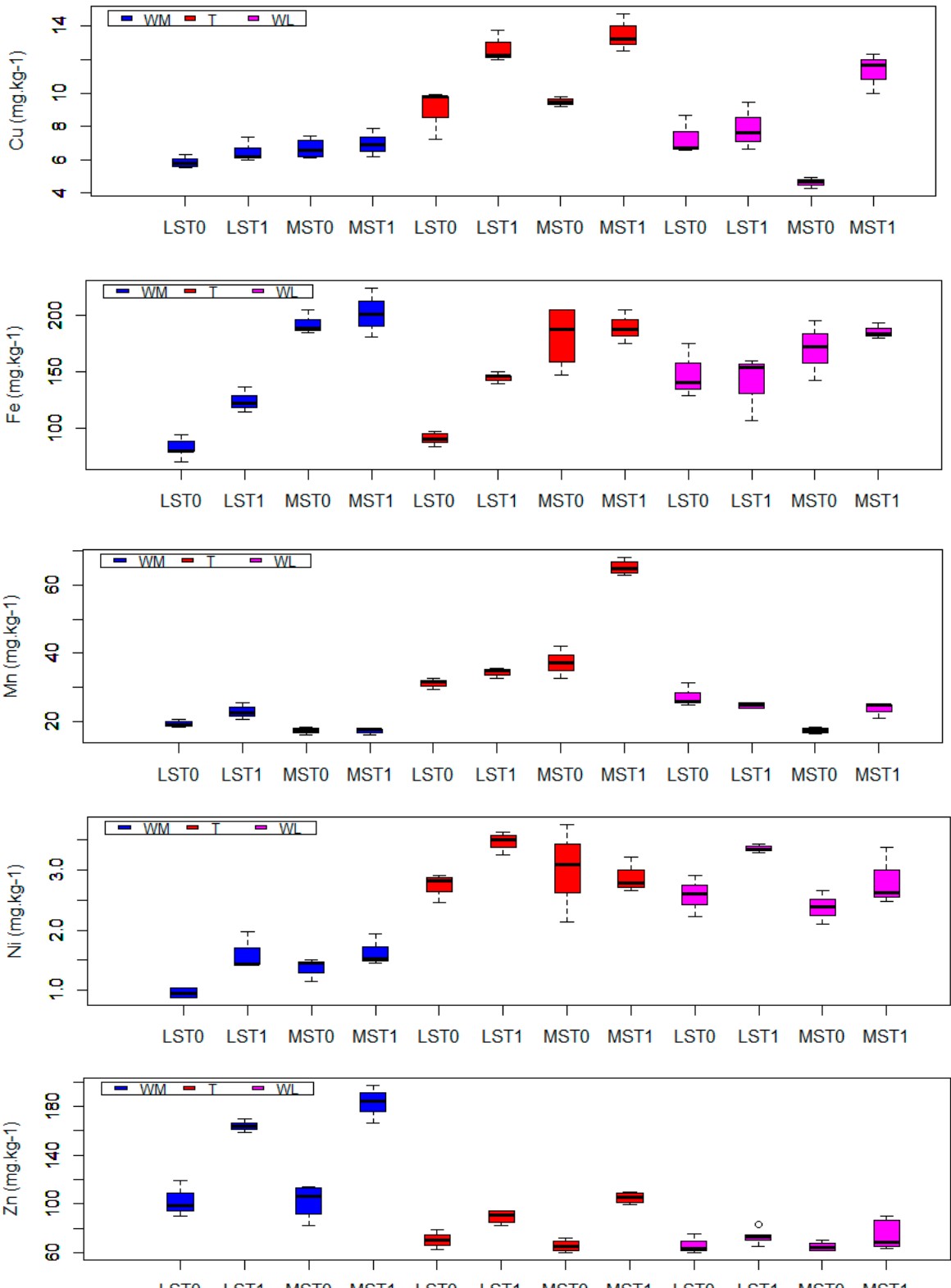

**Figure 2.** Content of microelements in grain. In the figure, the minimal value, first quartile (Q1), median, third quartile (Q3), and the maximal value are given to display the distribution of data. On the OX axis, the combination of treatments are marked, for more details see description in Material and Methods.

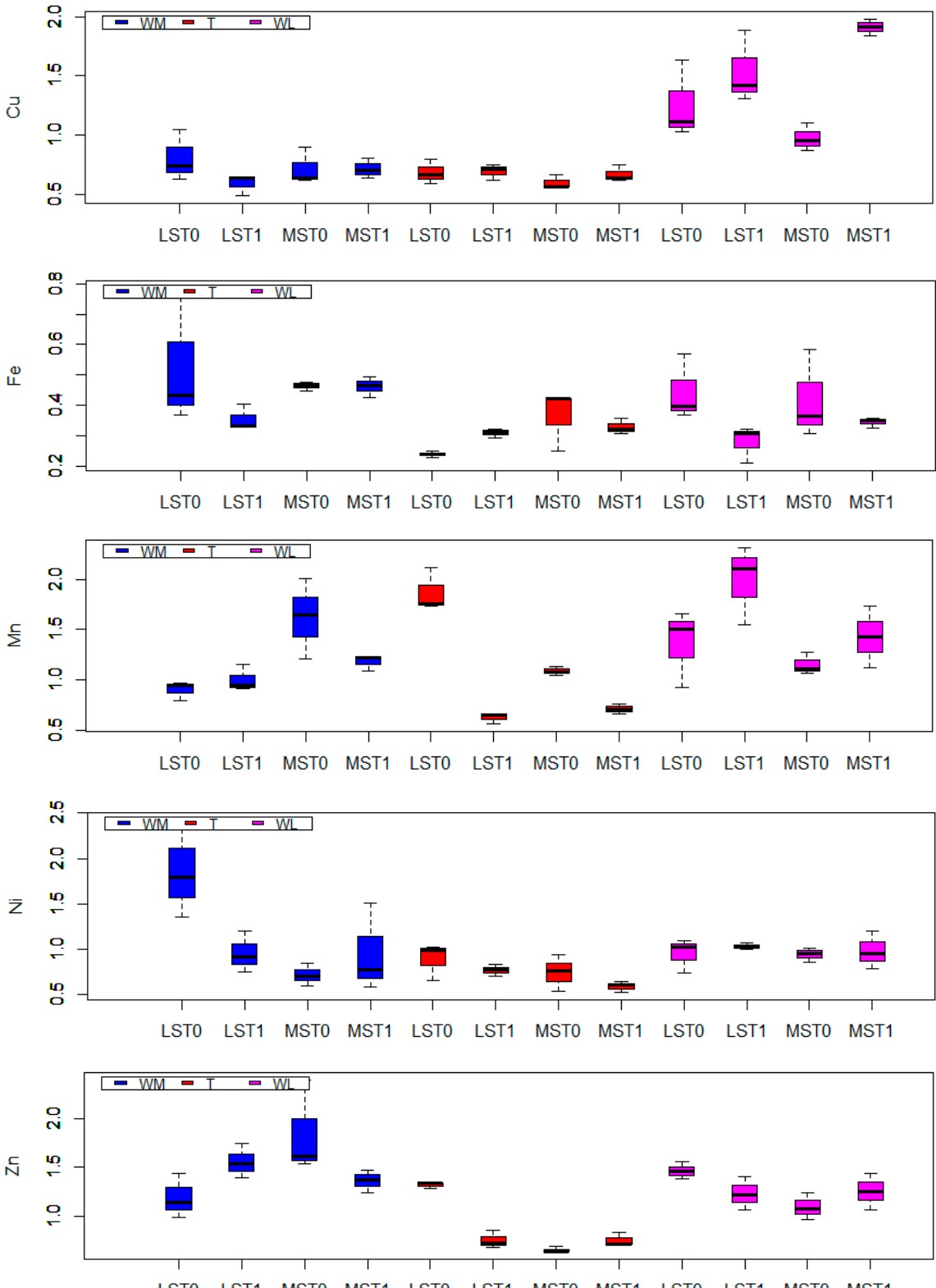

**Figure 3.** Harvest index (HI) values for plants (non-unit index). In the figure, the minimal value, first quartile (Q1), median, third quartile (Q3), and the maximal value are given to display the distribution of data. On the OX axis, the combination of treatments are marked, for more details see description in Material and Methods.

**Table 2.** ANOVA: empirical significance level p for studied: Zn, Cu, Mn, Fe and Ni contents.

|  | **Zn** | **Cu** | **Mn** | **Fe** | **Ni** |
|---|---|---|---|---|---|
| Shoots |  |  |  |  |  |
| A | 0.0000 | 0.0000 | 0.0000 | 0.0000 | 0.0000 |
| B | 0.0005 | 0.5931 | 0.0000 | 0.0000 | 0.0062 |
| C | 0.0000 | 0.0000 | 0.0000 | 0.0000 | 0.0000 |
| AxB | 0.8308 | 0.2045 | 0.0002 | 0.2838 | 0.0191 |
| AxC | 0.0000 | 0.0000 | 0.0000 | 0.1824 | 0.0759 |
| BxC | 0.0001 | 0.0010 | 0.0000 | 0.0001 | 0.0039 |
| AxBxC | 0.0000 | 0.0145 | 0.0316 | 0.8341 | 0.2787 |
| Grains |  |  |  |  |  |
| A | 0.0000 | 0.0000 | 0.1693 | 0.0002 | 0.0007 |
| B | 0.3881 | 0.7296 | 0.0013 | 0.0000 | 0.3740 |
| C | 0.0000 | 0.0000 | 0.0000 | 0.1933 | 0.0000 |
| AxB | 0.1425 | 0.0069 | 0.0001 | 0.8848 | 0.0340 |
| AxC | 0.0000 | 0.0008 | 0.1626 | 0.0292 | 0.5904 |
| BxC | 0.1194 | 0.3204 | 0.0000 | 0.0001 | 0.1567 |
| AxBxC | 0.2088 | 0.0007 | 0.0000 | 0.0811 | 0.6024 |
| HI |  |  |  |  |  |
| A | 0.0936 | 0.0034 | 0.0340 | 0.0480 | 0.1181 |
| B | 0.1535 | 0.9603 | 0.2310 | 0.2287 | 0.0063 |
| C | 0.0000 | 0.0000 | 0.0007 | 0.0035 | 0.0053 |
| AxB | 0.5836 | 0.0103 | 0.9646 | 0.4019 | 0.0453 |
| AxC | 0.3885 | 0.0000 | 0.0000 | 0.1899 | 0.1918 |
| BxC | 0.0035 | 0.7456 | 0.0002 | 0.7261 | 0.0404 |
| AxBxC | 0.0001 | 0.0956 | 0.0022 | 0.1783 | 0.0123 |

Factor A: compost addition (T0, T1), Factor B: soils (MS, LS), Factor C: plants (WM, T, WL). Significant differences at the level 0.05 are indicated in bold.

Similarly, a statistically significant interaction was found between compost and plants, since Zn, Cu and Mn contents increased in shoots of triticale cultivated on light and medium soil with SSC. The greatest dependence between soil and plant species was observed in the increased contents of all microelements in shoots of triticale cultivated on medium soil, regardless of compost addition (Table 2). The greatest interaction between Zn, Cu and Mn contents can be observed in the case of shoots of triticale cultivated on medium soil enriched with SSC, and it was manifested in greater contents of the above-mentioned micronutrients (Table 2, Figure 1).

Independently of the plant species and soil type, higher contents of all micronutrients with the exception of Ni were recorded in the grain of plants grown on soils enriched with SSC (Figure 2). Soil type had a significant impact on Mn and Fe contents in plant grain, and the contents of these two elements were statistically higher in plants grown on medium soil (Table 2, Figure 2). However, for the same environmental conditions, contents of other microelements in grain varied depending on the plant species. The highest contents of Cu and Mn were recorded in triticale grain. The amount of Zn in grain was significantly influenced by the interaction between compost application and cultivated plant species, as shown by the highest content of this microelement in grain of white mustard grown on SSC amended soils (Table 2, Figure 2). The content of Ni in grain was significantly affected only by the interaction between the compost and type of soil, and it was manifested by the highest Ni amount in grain of plants cultivated on soil of light agronomic class enriched with compost. The interaction between the compost application and type of soil had a significant impact on Cu, Ni and Mn contents in grain (Table 2), with their highest levels recorded in grain of plants grown on medium soil enriched in compost (Figure 2). Moreover, the interaction between the plant species and compost application had a significant effect on the contents of Cu, Zn and Fe. The highest level of Cu was recorded in grain of triticale, while for Fe, it was found in white mustard on soils amended with

SSC (Figure 2). The relationship between the type of soil and plant species was statistically significant in the case of Fe and Mn contents (Table 2), since the greatest content of Mn was observed in triticale grain, while the greatest content of Fe was observed in white mustard cultivated on medium soil. The interaction between the compost application, type of soil and plant species was manifested in the highest contents of Cu and Mn, determined in grain of triticale cultivated on soil of medium agronomic class supplemented with compost (Figure 2).

The SSC did not affect the harvest index assessed for Zn and Ni, while the HI calculated for Cu was the highest in the case of WL and soils enriched with compost, whereas the HI obtained for Mn and Fe was the highest for the control soil. Generally, the type of soil affected the harvest index value only for Ni, with the highest value recorded for light soil (Table 2, Figure 3). The crop plant species significantly modified the value of the harvest index. Greater HI values were recorded for Zn, Fe and Ni for white mustard, while for Cu and Mn the greater values were for white lupine. The simultaneous effect of compost application and type of soil had no significant impact on the harvest index calculated for Fe in all the cultivated plants (Table 2, Figure 3). The relationship between compost and soil determined for the harvest index was significant only in the case of Ni and Cu. Generally, the value of HI calculated for Cu was the highest for all plants growing on medium soil enriched with compost, whereas HI for Ni was the highest for all plants cultivated on light soil without compost addition. The relationship between the plant species and the presence of compost assessed for the harvest index was statistically non-significant for Zn, Fe and Ni, while it was significant in the case of the index determined for Cu and Mn in both soil types, with the highest values recorded for white lupine cultivated on soils enriched with compost (Table 2, Figure 3). The interaction between the soil type and plant species obtained for HI was statistically significant for Mn, Zn and Ni. It was manifested in higher HI values determined for Zn and Mn in the case of white mustard cultivated on medium soil without SSC and for Ni in white mustard cultivated on light soil. The interaction of all the three factors was statistically significant for HI values obtained for Zn, Mn and Ni (Table 2, Figure 3). The greatest effect of this interaction may be observed for HI values calculated for Zn in white mustard cultivated on medium soil without compost addition, for Mn in white lupine cultivated on light soil with SSC addition and for Ni in white mustard cultivated on light soil without compost addition.

This study analyzed the relationships between micronutrients accumulated in plants. Among cultivated crops only in the case of triticale no such relationships was confirmed statistically. Regardless of soil type and compost application, Fe amounts accumulated in the shoots were inversely proportional to Mn amounts (Table 3). A similar relationship was found for Zn and Fe in plants cultivated only on medium soil (Table 3).

Moreover, in white mustard the higher amounts of Mn were accumulated in shoots, the higher Zn level were recorded, and it was independent of experimental conditions (Table 3). Regression equations presenting the dependence of Ni in relation to Cu proved that Ni content in shoots of white lupine was directly proportional to Cu amounts, and this was regardless of soil type and compost application (Figure 3). On the other hand, a greater Cu accumulation in white lupine in shoots resulted in a lower Mn accumulation, and this was observed in the case of compost application (Figure 3). Regardless of soil type without compost, white mustard in grain accumulated less Mn, while simultaneously consuming more Ni (Table 3). The same antagonistic tendency between these elements was noted during fertilization, but it was not confirmed statistically.

**Table 3.** Coefficients in regression equation y= $b_0 + b_1$x and coefficients of determination (for regression coefficients statistically different from zero,$\alpha = 0.05$).

|  | Relation | Treatment | $R^2$ | $b_0$ | $b_1$ |
|---|---|---|---|---|---|
| Shoot | Fe(Mn) | WM LS T0 | 0.998 | 893.6 | −35.60 |
|  |  | WM LS T1 | 0.828 | 391.2 | −1.84 |
|  |  | WM MS T0 | 0.974 | 506.8 | −8.70 |
|  |  | WM MS T1 | 0.614 | 793.9 | −21.40 |
|  | Zn(Mn) | WM LS T0 | 0.929 | −100.8 | 10.30 |
|  |  | WM LS T1 | 0.943 | 59.8 | 2.08 |
|  |  | WM MS T0 | 0.942 | −8.6 | 6.30 |
|  |  | WM MS T1 | 0.948 | −728 | 56.80 |
|  | Ni(Cu) | WM LS T0 | 0.891 | 0.43 | 0.39 |
|  |  | WM LS T1 | 0.972 | −0.43 | 0.75 |
|  |  | WM MS T0 | 0.954 | −5.61 | 1.71 |
|  |  | WM MS T1 | 0.924 | 1.35 | 0.25 |
|  | Fe(Zn) | WM MS T0 | 0.809 | 739 | −1.32 |
|  |  | WM MS T1 | 0.914 | 494 | −1.40 |
|  | Mn(Cu) | WL LS T1 | 0.878 | 69.6 | −11.60 |
|  |  | WL MS T1 | 0.901 | 29.1 | −2.07 |
| Grain | Ni(Mn) | WM MS T0 | 0.941 | 2.37 | −0.14 |
|  |  | WL LS T0 | 0.926 | 1.98 | −0.12 |

## 4. Discussion

Micronutrients are elements that are essential for plants, but are required in small amounts. Deficiency of any of the micronutrients hampers normal plant growth, with plants exhibiting deficiency symptoms resulting in a reduction in yield and quality of crops [31]. Although the microelements are involved in all metabolic and cellular functions, plants differ in their requirements for these nutrients. Moreover, there is a significant variation between crop species in nutrient contents and their utilization, which results in their different nutrient requirements, and as a consequence, in the different chemical composition of plants. This is confirmed by the results obtained in this study (Figures 1 and 2). Regardless of the experimental conditions (soil type and compost application), cultivated plants accumulated the highest amounts of Fe and the lowest of Ni both in the grain and shoots. The order of micronutrients in terms of their amounts accumulated in plant organs was as follows: Fe > Zn > Mn > Cu > Ni. Tripathi et al. [1] analyzed the importance of micronutrients for plants, at the same time providing limit values for them. Comparing the data obtained in this work with those presented by the cited authors, it should be noted that the amounts of Cu and Zn were at a comparable level, while the tested plants in the experiment showed lower amounts of Fe and Mn, in comparison with the cited literature. Separately, the limit values for Ni were given by Kamboj et al. [32], and these thresholds were comparable to those obtained in this study.

Accumulation of micronutrients by plants is strictly related to their contents in soil and applied amendments. According to [33], regardless of soil type, the level of Fe was the highest and Ni the lowest. Additionally, composts, especially those prepared from sewage sludge, are characterized by higher amounts of Fe and lower of Ni [34]. Consequently, these factors jointly governed plant nutrient uptake by plants in relative amounts. Plants were found to vary markedly in their nutrient uptake levels. Based on mean values of microelement contents in the shoots and grain of plants cultivated in crop rotation, the highest amounts of micronutrients were usually accumulated in the following order: triticale > white lupine > white mustard. This could be related to individual plant requirements for nutrients, as well as the form and size of the root system. Frageria and Baligar [35] stated that the root system is strongly connected with the rate and pattern of nutrient uptake from soil. The high importance of roots for nutrient acquisition should be considered in two aspects, because specific root length is associated with high concentrations of nutrients, and it promotes their considerable acquisition capacity. On the other hand, a great root mass favors resource conservation [36]. For this

reason, both possibilities should be considered when interpreting the obtained results. Plants, such as white mustard or white lupine, having a vigorous and extensive root system, can theoretically explore large soil volumes, adsorbing more nutrients. Nevertheless, the present study does not indicate such a relationship, because triticale was characterized by higher amounts of all microelements in shoots and those of Cu, Mn and Ni in grain. Considering the content of the examined microelements in the studied parts of the plants, white mustard showed the lowest contents of Fe and Zn in shoots and Cu, Ni and Mn in grain. In turn, white lupine accumulated the lowest amounts of Cu, Zn and Mn in shoots and Zn in grain. Reliable explanations of these findings should be based on the slow mineralization process of SSC, which was thoroughly investigated and confirmed by [37,38]. In the first year of research, regardless of soil, WM showed the highest amounts of Fe and Zn both in shoots and grain. However, in the case of the second and third year of research, the contents of most microelements were higher in the organs of plants cultivated on soils enriched with compost, compared to those grown under control conditions. Another very significant mechanism is connected to micronutrient translocation and accumulation in plant tissues. The authors [39–41] stated that this mechanism varies for different elements and depends on their bioavailability. Statistical analysis of individual microelement acquisition in plant tissues proved it. Comparisons of ng the microelement amounts in the grain and shoots show one may state, that regardless of plants, their shoots accumulated more Fe and Ni, whereas grain were characterized by higher contents of Cu, Zn and Mn. Similar results were presented by [42]. On the other hand, the higher concentrations of Mn, Zn, Cu and Mo were found in the leaves and stalks of wheat, which those authors explained as founding part of their photosynthetic activity [41]. However, nutrients such as Cu, Zn and Mn are usually present at higher accumulations in the generative parts of plants. Authors [43] listed these elements, which were found as components in over 1500 proteins, whereas the largest group (>1200) is formed by Zn protein. According to the cited authors, proteins containing Fe, Cu or Mn make up groups of 50–150 members each, while for Ni it is only a few. This could be explained by the higher concentrations of Cu, Zn and Mn in grains of plants, which are the primary source of proteins. In contrast, Fe is involved in the formation of lignin and suberin in the plant cells [44], so it is mainly located in plant shoots.

A slightly different interpretation of obtained data may be provided by the harvest index. The mineral nutrient harvest index is a measure of translocation or utilization efficiency of the absorbed nutrients from plant vegetative parts to their grain [28]. This index may replace, and support, obtained results, because the lowest HI values were recorded for triticale and the highest for white mustard (especially in the case of Zn, Fe and Ni). In contrast to what had been expected, the harvest index data did not confirm the results presented above. This surprising finding may indicate insufficient nutrient utilization by triticale, in comparison to white mustard during the process of yield formation. The low HI values calculated for individual micronutrients may suggest limited remobilization and/or the importance of individual nutrients in grain production. In the present study the HI values differed widely between nutrients and plants, ranging from 0.75 to 1.56 (Zn), from 0.63 to 1.99 (Mn), from 0.59 to 1.91 (Cu), from 0.28 to 0.46 (Fe) and from 0.59 to 1.03 (Ni), respectively. The literature data present lower HI values for Cu, Zn and Fe [28] in relation to those shown above. Nevertheless, at the same time, the HI values calculated for Fe and Ni confirmed their lesser role in the production of grain proteins, and a far greater role for Cu, Zn and Mn.

The conducted statistical analysis indicates a significant effect of compost on the higher acquisition of individual elements in the analyzed plant parts, and a simultaneous lack of any significant effect of SSC on the harvest index. Generally, micronutrient amounts assessed both in the shoots and grains of plants cultivated in soils amended with SSC were higher, in comparison to their relative amounts in plants grown in soils without compost application. These results are similar to those of [21], because in the cited study, compost applied to the soil increased the concentrations of Cu and Zn in plant leaves. Additionally, a study conducted by [41] indicates a significant influence of organic fertilizer on enriching plant leaves and stalks with Mn and Cu, in comparison to those grown in the conventional farming system. The opposite findings concerning organic fertilization were presented

in a study conducted by [17], where the incorporation of manure to soils reduced the availability of micronutrients, and consequently decreased their uptake by plants. However, the authors showed that soil type had a significant effect on Cu, Zn and Mn uptake by plants, because concentrations of Cu, Zn and Mn in leaves were higher on sandy soil. In the presented work, a significant effect of medium soil was obtained in the case of microelement contents accumulated in plant tissues. Contradictory results are provided by statistical analysis performed for HI, because neither soil type nor compost application had a significant effect on the HI values. These findings partly explain the weak soil effect, because mean amounts of nutrients determined for individual plants (jointly for shoots and grain) cultivated on light and medium soil were comparable and irrespective of SSC application they ranged from 87.4 to 94.8 mg·kg$^{-1}$ (Zn), from 244.9 to 347.2 mg·kg$^{-1}$ (Fe), from 22.5 to 26.6 mg·kg$^{-1}$ (Mn), from 9.5 to 9.7 mg·kg$^{-1}$ (Cu) and from 2.6 to 2.7 mg·kg$^{-1}$ (Ni), respectively.

One of the hypotheses presented in this study was that, depending on the type of soil fertilized with compost, the differences would be observed in the nutrient uptake by plants. Such an assumption is connected not only to commonly known differences in the physicochemical and chemical properties of soils, but also related to this various rate of transformation in the introduced organic matter, e.g., with compost. Referring to the literature [45–47], since the SSC mineralization process mainly depends on soil texture, moisture regime, microbial activity and the quantity of organic matter incorporated into soil, a more rapid decomposition and incorporation of nutrients may be assumed in light soil because of more favorable conditions, which may accelerate this process relatively soon after compost application. Coarse textured soils such as, e.g., light soil are characterized by a lower water-holding capacity and good aeration. Fine-textured soils, e.g., the medium one, have different conditions. Physical properties are more favorable in light soil, and they may accelerate the mineralization process at an early period after compost application [13]. However, the obtained results did not support these assumptions, which was indicated by small differences in nutrient concentrations in the analyzed plant parts, as confirmed by statistical analysis (Table 2). Nevertheless, the influence of the interaction between SSC and soil type on nutrient accumulation in shoots and grain of plants was mostly in medium rather than in light soil (Figures 1 and 2). The lack of a strong effect of SSC in individual soils could be explained by their satisfactory fertility expressed in adequate levels of available macronutrients. Additionally, Dada et al. [20] stated that compost activity in supplying nutrients for plants is more effective in deficient soils. Additionally, the results obtained by [48] revealed that winery solid waste composts could serve as potential good sources of K and Zn for maize production, particularly in sandy soils where these nutrients are often reported to be deficient.

It is necessary to emphasize the multidimensional nature of the presented research, which may constitute an interpretative challenge connected with many experimental variables. Thus, the employed proper statistical methods may apparently facilitate interpretation and explanation of the findings. The statistical analysis based on three-way ANOVA is indicated in the literature as a validated statistical tool [49,50]. Performed analysis allowed the correct and clear assessment of the obtained data, and its usefulness in explaining natural phenomena should be appreciated.

## 5. Conclusions

Differences were found between plants varying in nutrient requirements, cultivated in crop rotation under similar environmental conditions and in terms of their acquisition efficiency. The distribution of nutrients in plant parts (shoots and grain) reflects their metabolic and cellular functions in crops, which was particularly evident for Cu, Zn, Mn and Fe. In general sewage sludge, the compost application significantly increased the concentrations of Cu, Zn and Mn in the grain of cultivated plants. At the same time, the shoots of plants were enriched with Fe and Ni. The obtained results indirectly indicate that SSC used in agricultural practice turned out both to be a slowly decomposed material and a valuable source of micronutrients. In comparison to both crops grown in the first year under control conditions, plants fertilized with SSC were characterized by a higher content of microelements, as was found in the case of the triticale and white lupines cultivated in the second and third year of the

experiment. Additionally, the calculated harvest index values showed poor utilization of microelements by triticale, and far better utilization by white mustard. The harvest index may be considered a useful tool for the assessment of micronutrient translocation within plant organs. Additionally, the effect of sewage sludge compost on the chemical composition of plants cultivated in crop rotation was more statistically significant in medium soil.

**Supplementary Materials:** The following are available online at http://www.mdpi.com/2073-4395/10/5/642/s1, Figure S1. Climate graphs characterizing weather conditions in Swojec.

**Author Contributions:** Conceptualization, M.J.; investigation, M.J.; methodology, M.J., M.G.; project administration, M.J.; data curation, M.J., M.G.; writing –original draft, M.J., M.G.; writing –review and editing, M.J., M.G.; funding acquisition, M.J. All authors have read and agreed to the published version of the manuscript.

**Funding:** The publication was co-financed within the framework of Ministry of Science and Higher Education programme as "Regional Initiative Excellence" in years 2019-2022, Project No. 005/RID/2018/19".

**Conflicts of Interest:** No potential conflict of interest was reported by the authors.

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
