# Peer review of "Microelement Variability in Plants as an Effect of Sewage Sludge Compost Application Assessed by Different Statistical Methods"

_agronomy, doi:10.3390/agronomy10050642_

Round 1

Reviewer 1 Report

In my opinion, the manuscript is well structured and the results adequately explained. However, I find some major concerns that should be properly assessed prior to its consideration for publication.

The authors do not consider the negative drawbacks that could be inherent to sewage sludge compost aplication. There are safety thresholds that could limitate the utilization of this amendement in agricultural soils:

  • EC, 2001. Working document. Biological treament of biowaste (2nd draft)
  • EC, 2001. Commission Regulation (EC) no. 466/2001 of 8 March 2001 setting maximum levels for certain contaminants in foodstuffs. Official Journal of the European Communities, 16.3.2001, L77/1-23

It is known that not only micronutrient deficit has a detrimental effect on plant nutrion, as their optimum ranges are characterized to be in a very narrow scale. I miss information about micronutrient limits and the adverse effects associated to their excesive concentration in plants.

I also find that the experimental design is not the best suited to validate the second hipothesis about SSC mineralisation rate. The authors only measured nutrient concentrations in plant shoots and grains, and plant species were rotated (so, comparisons between different cropping seasons is biased). At least, soil mineral nutrient content should has been mesured to be able to answer this research question.

I suggest the authors to give more focus to their own results in the discussion section (there is much more information about previous research)

I suggest to homogenise figure 3 with the other figures and inlude in the y axis the elements (and units). Figures and tables footnotes should be more explicative, all of them lack in the explanation of the abbreviations and content (what is it represented in the boxes, error bars, etc.?). I also suggest to include the cropping periods in the figures (1st, 2nd and 3rd year)

Other comments are provided in the attached document

Reviewer 2 Report

This paper investigates the effects of soil amendment with sewage sludge compost (SSC) on the microelements accumulation in plants. This is a very interesting topic that take into consideration plant nutrition from the point of view of circular economy. But there are some weak points through the paper as follows:

  1. Introduction doesn’t analyse properly and satisfactory the state of art (a lot of reference are missing in supporting the sentences and topics presented along the text).
  2. Mat & met - The experimental design is not clear to the reader. Considering the complexity of the  trial, I would suggest to describe better the experimental design (control, number of treatments, number of plant tissues, samples, replicates…)
  3. Results – Figures cited along the text are not in the manuscript. It is difficult to
  4. Discussion – authors only provide descriptive information and it is not always clear and speculative in some parts. They should examine in depth the multiple mechanisms involved.
  5. General consideration - In order to use safety sewage sludge compost in agriculture, it is extremely important to know its quality. Did the authors performe analysis of phytotoxicity, microbial load, and heavy metal content in order to verify they respect the standard limits for agricultural use?

For these reasons, my opinion is that the paper is not suitable for publication as it is, but it needs major revision. I wrote some specific comments below, in the hope that they will be useful to the authors.

Line 41  […] environmental stress conditions (add ref.)

Line 45 […]  the most crops are very susceptible to micronutrient deficiency (add ref.).

Line 50 “Unfortunately these fertiliser are costly.” Please add a sentence on the importance of have a more sustainable management and on the increasing attention of consumer to healty for human and environment.

Line 53 add ref.

Line 63: from environmental point of view, the authors could also cite the recycle and reuse of organic matrix as nutrient source.

Line 66: please delete “see”

Line 66-74 In order to better describe the scenario under which the experimental trial was performed, I would suggest to move this part at the beginning of the Introduction.    

Line 76 “slow mineralization” is also linked to several factors like temperature... I would suggest to add a sentence and some ref about this.

Line 87-89 Please re-write the sentence in a more clear and correct way.

Line 106: In my opinion the experimental design description is needed in the text of mat & met and not only citing a ref.

Line 112: please replace Mg with t (tonne)

Line 114: T1: add information about the time after T0 (days…)

Line 120-122 Please describe and add information about the Polish weather conditions (rainfall, min and max temperature, Et0, VPD…). Authors could also document  the weather conditions with graphs as Supplementary materials.

Line 125 please simply describe, even not in details, the method you used. It is not easy to read the paper if mat & met often refers to other ref.

Line 124-125 This should be moved before describing plants in the experiment, after line 117.

Line 137: furnace: please add the model and manufacturer.

Line 137-138; 142-143 please move the ref at the end of sentence.

Line 173-174 “control soil vs soil enriched in compost”: in Mat & met is not clear the different samples and the comparison between treatments. Please clarify.

Line 188, Line 200: Figures 1-3 seem to be missing along the paper.

Results should be integrated with values of micronutrients content in shoot and grain and dry or fresh weight of sampled plant tissues.

Line 267 “there is a significant variation both between crop species and genotypes of the same species”. The differences among different genotypes of the same species is not among the results of the paper. I suggest to delete it.

Line 276 Synergistic and antagonistic effect between nutrients. This is the first time that it is mentioned. I suggest to add some sentence on relationship between nutrients in the introduction.

Line 288, 294-295: “form and size of root system”:  Did you measured the root biomass, root density or other measures related to the root system? Have you got some photos of root systems of different crops? Without any data on roots the sentence seems to be speculative.

Line 315-317: “The distribution of nutrients in plant parts (shoots and grains) reflects differences of accumulation between different plant tissues.” The long-distance transport of micronutrients is a very complex mechanism and it seem to be described in too much simplicistic way. Please better explain the mobility of nutrients in plant system.

Round 2

Reviewer 1 Report

Some minor comments are displayed in the attached document

Author Response

Dear Editor,

Thank you very much for additional referee’s comments. After re-review, the new version of the work includes all comments indicated by the reviewers. And so Mg was changed to t, the descriptions of micronutrients was uniformed, the quality of figures and their descriptions  were improved. In relation this additional information was enclosed in material and Methods chapter. The conclusions were shortened and minor comments were made in the text as suggested by the reviewer. Doi numbers will be introduced when we will get clear information about this from the publisher.

Reviewer 2 Report

Dear Editor and authors,

the new version of the manuscript has been improved, but it needs some more changes before publication. I would like to add few comments as follows, hoping that they could be useful:

-line 23: please replace Mg with t

-Figures with their captions should be clear to the reader even without reading the manuscript. Captions of figures 1-3 need to be improved, with a more clear description of all the acronyms. 

-please be consistent with the name of elements (e.g. Zn or zinc): along the text they are both (e.g. line 335, 351...), please choose to use one. 

Author Response

(The authors gave the same response as above.)
